# From Aggregation to Guidance: Strategies for Personalized Federated Fine-Tuning of Foundation Models

**Mikołaj Piórczyński***   **Wojciech Łapacz***   **Xinyu Li**   **Chang Liu**
**Abby Turner**   **Artur Dubrawski**
Auton Lab, School of Computer Science, Carnegie Mellon University
{mpiorczy, wlapacz, xinyul2}@andrew.cmu.edu

## Abstract

Federated learning (FL) enables collaborative fine-tuning of pretrained foundation models in privacy-sensitive settings without directly sharing raw data. Personalized federated learning (PFL) further addresses client-side heterogeneity by learning models tailored to each client's local tasks while still benefiting from cross-client collaboration. In this work, we study strategies to improve the effectiveness of personalized federated fine-tuning of Large Language Models (LLMs) using LoRA. On the server side, we explore advanced aggregation strategies that go beyond simple parameter averaging, drawing on model merging techniques to construct a robust global model. On the client side, we investigate different ways to leverage the global model to guide local learning, including hard initialization, parameter regularization, and function-space guidance via knowledge distillation, and propose a similarity-based adjustment strategy to further improve local learning. Empirical results on the Super NaturalInstructions dataset demonstrate that careful design of both server- and client-side strategies has the potential to improve PFL performance, providing insights for developing more effective PFL learning frameworks for fine-tuning LLMs for heterogeneous client tasks.

## 1 Introduction

Large pretrained foundation models, such as Large Language Models (LLMs), have demonstrated strong performance across a wide range of downstream tasks, and fine-tuning them on specific datasets further improves their effectiveness in real-world applications. However, collecting large-scale fine-tuning datasets is challenging due to high costs and privacy concerns. Federated learning (FL) [34] has emerged as a promising solution for collaborative fine-tuning of foundation models, as it enables the use of client-side data without directly sharing it, making it particularly appealing in privacy-sensitive domains such as healthcare [2] and finance [17]. To further reduce computational and communication overhead, recent studies often adopt parameter-efficient fine-tuning (PEFT) techniques, particularly Low-Rank Adaptation (LoRA) [9], for federated LLM fine-tuning [34, 1, 31].

*Personalized Federated Learning* (PFL) [24] extends standard FL to address the additional challenge of heterogeneity of client-side tasks, where clients focus on different local tasks and their data no longer follow a similar distribution. Unlike in standard FL where clients collaboratively train a single global model, in PFL each client learns a model tailored to their local task, while still leveraging the shared knowledge captured by a global model trained through cross-client collaboration.

The effectiveness of PFL depends on two key components: (1) **Server-side aggregation:** how to aggregate client models into a robust global model that benefits all clients, and (2) **Client-side**

---

*Equal contribution.

**learning guidance:** how to effectively leverage the shared knowledge captured by the global model to guide each client's local training and improve performance on their specific task. Standard FL aggregation methods such as FedAvg [21] are known to be less effective under task heterogeneity, as conflicting update directions from diverse data distributions can lead to degraded performance and slow convergence [36]. Consequently, initializing each client from the global model at the start of each round may not provide an optimal starting point for subsequent local training.

In this paper, we study different strategies for both components to improve the overall effectiveness of personalized federated fine-tuning of foundation models with LoRA. For server-side aggregation, we draw on advanced *model merging* methods that go beyond simple averaging to more effectively unify client models into a robust global model. For client-side learning, we investigate three approaches to leveraging the global model to guide local training—hard initialization, parametric regularization, and function-space regularization—and propose similarity-based regularization adjustment strategies to improve local learning. Our empirical results demonstrate that advanced model merging, regularization-based guidance, and similarity-based adjustment together can improve the overall effectiveness of personalized federated fine-tuning of foundation models.

## 2 Preliminaries

We consider a personalized federated learning (PFL) setup with LoRA fine-tuning [31]. There are $K$ clients, each hosting the same frozen pre-trained LLM $W_0$ and its own local dataset $D_1, \ldots, D_K$, and trainable LoRA parameters $\Delta W_1 = B_1 A_1, \ldots, \Delta W_K = B_K A_K$. The goal is to collaboratively learn personalized LoRA parameters for each client by optimizing the sum of client objectives $\mathcal{L}_k$:

$$\min_{(A_1, B_1), \ldots, (A_K, B_K)} \sum_{k=1}^{K} \mathcal{L}_k(D_k; A_k, B_k).$$

In each communication round $t$, the server aggregates local updates from the clients to produce a global model $\Delta W_G = B_G A_G$, for client use at the beginning of the next round. $A_G^{(t)}, B_G^{(t)} = \mathcal{M}(\{A_k^{(t)}, B_k^{(t)}\}_{k=1}^{K})$, where $\mathcal{M}(\cdot)$ denotes the merging function, e.g., simple averaging in FedAvg. We explore alternative merging strategies in Section 3.

After receiving the global model, clients can incorporate it into their next local training round in different ways—e.g., by initializing from the global model or by using it to regularize local parameter updates. We investigate such strategies in Section 4.

## 3 Server-side Aggregation

In this section, we study the server-side construction of global models using advanced model merging methods that go beyond simple parameter averaging. *Model merging* methods typically take the general form $W_M = W_0 + \alpha \mathcal{M}(\{\Delta W_k\}_{k=1}^{K})$, where $W_0$ is the pre-trained model, $\Delta W_k = W_k - W_0$ denotes the client-specific fine-tuning updates, and $\alpha \geq 0$ is a scaling coefficient. For LoRA-based fine-tuning, $\Delta W_k = B_k A_k$ where $A_k, B_k$ are the low-rank LoRA matrices.

We experiment with two model-merging methods: (1) **TIES-Merging [30]:** mitigates parameter-level interference by pruning insignificant updates, resolving sign conflicts, and merging only aligned directions across clients, and (2) **Iso-C [19]:** performs isotropic merging by flattening the singular value spectrum of the merged weight matrix, thereby reducing the dominance of a few top directions and improving the alignment of client models with the global model. Both methods can mitigate the conflicts across individual models and produce a more robust merged model than simple averaging.

Additionally, we experiment with two approaches for aggregating client updates: (1) **A & B:** aggregation is performed separately on the LoRA matrices $A_k$ and $B_k$, as in FedIT [34], and (2) **KnOTS [22]:** aligns task-specific models within a shared subspace using stack-SVD, which has shown superior effectiveness in merging LoRA fine-tuned models. Concretely, stacked client updates are decomposed as $[\Delta W_1^{(t)}, \ldots, \Delta W_K^{(t)}] = U\Sigma[V_1^\top, \ldots, V_K^\top]$ where $U\Sigma$ forms the shared subspace. The global model is then obtained by $\Delta W_G^{(t)} = \alpha U\Sigma \mathcal{M}(\{V_k^\top\}_{k=1}^{K})$. Finally, we apply truncated SVD (TSVD) to recover the low-rank global LoRA matrices $A_G^{(t)}, B_G^{(t)}$ from $\Delta W_G^{(t)}$, following FlexLoRA [1]. For both approaches, we also experiment with the ReLoRA strategy [15], which incorporates the full

Table 1: Mean ROUGE-L across clients under different merging strategies for global model construction (higher is better).

| Merging Method | ReLoRA | ROUGE-L ($\uparrow$) | |
|---|---|---|---|
| | | A & B | KnOTS |
| FedAvg | No | 46.70 | 42.82 |
| | Yes | 37.24 | 39.39 |
| TIES-Merging | No | 40.02 | 35.79 |
| | Yes | 39.02 | 40.77 |
| Iso-C | No | 43.38 | **48.31** |
| | Yes | 36.57 | 45.16 |

merged global model $\Delta W_G^{(t)}$ into each client's backbone and then reinitializes the client-specific LoRA adapters from scratch.

We conduct experiments on the Super NaturalInstructions dataset [27], which contains a wide variety of natural language generation (NLG) tasks. To simulate a task-heterogeneous FL setting, we select the 10 tasks with the largest sample sizes and assign each to a distinct client. For computational efficiency, we cap each client's training split at 500 samples and the test split at 50 samples. We use LLaMA-3.2-1B [7] as the base LLM, with each client model fine-tuned using LoRA of rank 8 applied to the Q and V projection matrices in the attention layers. Training is conducted for 20 communication rounds, where in each round, clients train locally for 1 epoch with batch size 32 and learning rate $5e-5$ using the AdamW optimizer. For evaluation, we report the average ROUGE-L score across all clients, computed on each client's local test set using their local model. The global model merging coefficient $\alpha$ is tuned through hyperparameter search on a held-out validation set.

As shown in Table 1, Iso-C merging in KnOTS space with TSVD (without ReLoRA) achieves the best performance on client tasks. This is not suprising as KnOTS mitigates client parameter conflicts by aligning updates in a shared subspace, while Iso-C further reduces interference by flattening dominant directions, preventing the global model from being biased toward any single client. FedAvg in the A & B space yields competitive results. In constrast, TIES-Merging underperforms in most configurations. Since TIES-Merging includes a pruning step, this suggests that pruning partially converged models—especially in the PFL setting—may hinder convergence and final performance. Interestingly, applying ReLoRA in the KnOTS space improves TIES-Merging, indicating that fully incorporating unified global representations into the backbone while maintaining client-specific adaptation through fresh LoRA initialization can be an effective strategy for PFL. Overall, our preliminary experiments highlight the promise of leveraging advanced model merging strategies for improving personalized federated fine-tuning of LLMs with LoRA.

## 4   Client-side Learning Guidance

In this section, we investigate different strategies to use the global model to guide local client training, and propose a similarity-based adjustment strategy to improve client-side training. We focus on three common approaches: (1) **hard initialization**, where client models are directly initialized from the global model, (2) **parametric regularization**, where the global model acts as a regularizer for client parameter updates, and (3) **function-space regularization**, where the global model serves as a teacher and guides client models through knowledge distillation. Hard initialization amounts to minimizing the supervised fine-tuning loss $\mathcal{L}_{SFT}$ starting from initialization, while the latter two approaches minimize $\mathcal{L} = (1-\beta)\mathcal{L}_{SFT} + \beta\mathcal{L}_{REG}$, with $\mathcal{L}_{REG}$ defined according to the chosen approach, and $\beta$ is a scaling coefficient to control the regularization strength.

For parametric regularization, $\mathcal{L}_{REG}$ is defined as an L2 penalty on the LoRA parameters, i.e. $\mathcal{L}_{REG} = ||A_k^{(t+1)} - A_G^{(t)}||^2 + ||B_k^{(t+1)} - B_G^{(t)}||^2$. In other FL settings, such penalties have proven effective in mitigating the negative effects of client heterogeneity [13]. Intuitively, guiding client updates throughout training helps keep them collocated, which can improve the effectiveness of server-side aggregation and produce a more robust global model for subsequent client-side training.

Although appealingly simple and efficient—with computational costs proportional to LoRA rank—distance in parameter space may not faithfully reflect function-space distances between models. In our setting, scaling $A_k^{t+1}$ and $B_k^{t+1}$ by $\frac{1}{c}$ and $c$ leaves the underlying adapter unchanged, yet results in different penalty values. To address this, we also experiment with function-space regularization

through knowledge distillation (KD) [12, 36], by introducing a KL divergence regularizer that aligns the outputs of the client model and the global model on local data. Let $p_k(\cdot|x, T)$ and $p_G(\cdot|x, T)$ denote local and global output probabilities over the vocabulary $V$ given input text $x$, obtained by applying a softmax with temperature $T$ to the output logits. Given a batch of local data $D = \{(x_k, y_k)\}$, with $L(y_k)$ denoting the length of $y_k$ and $(y_k)_{\leq i}$ denoting the prefix of length $i$, the function-space regularizer takes the form

$$\mathcal{L}_{REG} = \frac{T^2}{|V||D|} \sum_{(x_k, y_k) \in D} \frac{1}{L(y_k)} \sum_{i=1}^{L(y_k)} \sum_{v \in V} p_k(v|(x, (y_k)_{\leq i}), T) \log \frac{p_k(v|(x, (y_k)_{\leq i}), T)}{p_G(v|(x, (y_k)_{\leq i}), T)}.$$

We adopt the same experiment setup as in Section 3, using the same 10 clients simulated from the Super NaturalInstructions dataset. Table 2 shows the best FedAvg and Iso-C results obtained with hard initialization (as in Section 3), alongside the results achieved using parametric regularization and function-space regularization for client-side guidance.

Table 2: Mean ROUGE-L across clients for different guidance strategies (higher is better).

| **(a) FedAvg** | | | **(b) Iso-C** | |
|---|---|---|---|---|
| **Method** | **ROUGE-L ($\uparrow$)** | | **Method** | **ROUGE-L ($\uparrow$)** |
| Best Hard-init | 46.70 | | Best Hard-init | 48.31 |
| Parametric Reg. | **52.05** | | Parametric Reg. | 52.54 |
| Function-space Reg. | 51.67 | | Function-space Reg. | **52.96** |

For both merging strategies, parametric and function-space regularization outperform hard initialization. The improvements are comparable, despite the apparent differences between the two approaches. This may be explained by the fact that a second-order Taylor approximation of the KL divergence corresponds to a Fisher-weighted parameter distance [3], which could potentially account for the similar gains achieved by both methods.

Lastly, we note that uniform regularization across clients may be insufficient to achieve optimal PFL performance. For instance, clients whose tasks differ significantly from the rest of the federation may benefit from reduced regularization toward the global model. To address this, we propose to leverage the similarity between the global and client models to modulate the regularization strength. Specifically, we weight the regularization penalty $\beta$ based on a similarity score, and experiment with two measures: (1) cosine similarity between flattened and concatenated LoRA matrices, and (2) subspace alignment ratio (SAR) proposed by [19], which measures model alignment via the subspace spanned by the top left singular vectors of the weight matrices. Table 3 shows the Iso-C and FedAvg results using their respective best guidance strategies, compared to the same strategies with similarity-weighted adjustment.

Table 3: Mean ROUGE-L across clients for different similarity-weighted adjustment (higher is better).

| **(a) FedAvg (w/ Parametric Reg.)** | | | **(b) Iso-C (w/ Function-space Reg.)** | |
|---|---|---|---|---|
| **Method** | **ROUGE-L ($\uparrow$)** | | **Method** | **ROUGE-L ($\uparrow$)** |
| Fixed $\beta$ | 52.05 | | Fixed $\beta$ | 52.96 |
| Cos Sim $\beta$ adj. | 52.37 | | Cos Sim $\beta$ adj. | **53.66** |
| SAR $\beta$ adj. | **52.71** | | SAR $\beta$ adj. | 52.34 |

We observe that adjusting regularization strength on a per-client basis can improve PFL performance. Specifically, SAR-based adjustment yields the best results for FedAvg, while cosine similarity performs best for Iso-C. This difference may arise as Iso-C already increases the SAR between global and client models during server-side aggregation, whereas FedAvg can benefit more from explicitly leveraging SAR for modulating regularization. We leave a systematic study of how model alignment can be leveraged to further improve local learning as an important direction for future work.

**Conclusion**    In this paper, we study personalized federated fine-tuning of foundation models with LoRA under client heterogeneity, investigating different strategies on both the server and client sides. Our findings show that (1) advanced model merging methods have the potential to improve the global

model in ways that benefit local client tasks, (2) parametric and function-space regularization can effectively leverage the global model to guide client training, and (3) our proposed similarity-based adjustment strategy can further improve the performance.

## Acknowledgments and Disclosure of Funding

This work has been partially supported by the National Science Foundation (awards 2427948, 2406231 and 2530752) and Defense Advanced Research Projects Agency (award HR00112420329).

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

## A  Related Works

**Model merging** refers to the process of combining multiple independently trained models into a single unified multi-task model, typically without additional fine-tuning. Most of recent model merging techniques build upon the concept of *Linear Mode Connectivity* (LMC) [6, 5], which observes that models fine-tuned on related tasks often lie within connected low-loss regions of the parameter space. Leveraging this, Task Arithmetic [10] proposes linearly combining task-specific deltas. As noted in [26], the setup and success of Task Arithmetic is thus quite similar to that of classic Federated averaging in a one-shot federated learning setup [8]. Fisher merging [20] treats parameters as Gaussian posterior samples, weighting them by Fisher information for improved robustness over naive averaging. Recently [3] explored Fisher merging [20] for federated learning, but from the perspective of function space aggregation, showing that aggregation respecting functional representations of models can outperform those treating only parametric representations.

More advanced techniques aim to reduce task interference during merging. TIES [30] and DARE [33] mitigate conflicts between task-specific updates by pruning insignificant changes, resolving sign conflicts, and aligning update directions before merging. Iso-C [19] further refines this approach by addressing geometric misalignment in weight space. It flattens the singular value spectrum of the merged model to improve alignment between individual task-specific models and the resulting merged model. However, [25] observe that standard model merging methods encounter significant challenges when applied to LoRA-finetuned models, since LoRA-adapted models are less aligned comparing to their fully fine-tuned counterparts. To that end KnOTS [22] leverages singular value decomposition (SVD) to jointly transform the weights of different LoRA models into a shared, aligned space, where traditional merging techniques can be more effectively applied.

**Personalized Federated Learning** (PFL) extends the traditional Federated Learning (FL) paradigm by enabling each client to train a model tailored to its unique data distribution. Classic FL, e.g. with federated averaging or similar strategies adapted to LLM tuning such as FedIT [34], effectively pursues a single model trained on the global data distribution. It is well known in FL that these methods often struggle to learn in the presence of heterogeneous data [14]. While strategies employing additional regularization [13] or replacing parametric aggregation with aggregation in function spaces [16] can help alleviate convergence issues, PFL has grown from the recognition that enforcing a single shared global model for distributed learning and LoRA tuning is often insufficient in practice.

Thus PFL methods seek a balance between collaboration and personalization, leveraging the pool of global data but allowing clients further specialization to their local distributions. Numerous techniques have been proposed to achieve this balance, including model decomposition into shared and personalized components [28, 18, 35], regularization-based approaches respecting this decomposition [23, 32], meta-learning [4], and debiasing client objectives [11]. Limitations of the global model mean that these methods may still struggle to retain task performance in heterogenous settings,

motivating strategies in [29] to enforce separation between parametric representations of adapters from different tasks. While this is done throughout learning rather than post hoc, the similarity to challenges in model merging motivates us to investigate how merging techniques can be employed to improve performance in PFL pipelines.

