# OpenReview forum: "From Aggregation to Guidance: Strategies for Personalized Federated Fine-Tuning of Foundation Models"
_NeurIPS.cc/2025/Workshop/UniReps — UniReps2025_

### Official Review · Reviewer_1nkH · 2025-09-12
**The work studies different strategies for personalized federated LoRA fine-tuning in LLMs, but needs broader methods and evaluations.**

**Confidence:** 4

**Review:**

The work explores different strategies for server-side aggregation and client-side learning in order to more effectively leverage the shared knowledge encoded in the global model. Specifically, it studies how these strategies impact personalized federated LoRA fine-tuning in LLMs. The paper provides insights into the design of improved algorithms for balancing personalization with global generalization.

That said, there is room for a broader and more systematic evaluation. In particular, the study would benefit from exploring a wider range of aggregation methods for instance, FedEx-LoRA [1], FFA-LoRA [2], and other emerging approaches, to better capture the trade-offs between efficiency, personalization, and robustness. Similarly, expanding the evaluation to cover more model families and diverse tasks would strengthen the generality of the conclusions and provide a clearer picture of how the proposed strategies scale across different settings.

Overall, the work makes a contribution by highlighting the role of aggregation design in personalized federated LoRA fine-tuning, while also pointing to several promising directions for detailed empirical validation.

[1] FedEx-LoRA (ACL 2025 Oral): https://arxiv.org/abs/2410.09432

[2] FFA-LoRA (NeuRIPS 2024): https://arxiv.org/abs/2403.12313

**Score:**

3

**Topic Fit:**

2

---

### Official Review · Reviewer_ihaC · 2025-09-12
**Interesting Heuristic for Personalized FL, but Limited Novelty and Weak Experiments**

**Confidence:** 4

**Review:**

## Summary
This paper studies strategies for **personalized federated fine-tuning of LLMs with LoRA**. It examines both **server-side aggregation** and **client-side guidance**. On the server side, the authors adapt **model merging techniques** (e.g., TIES, Iso-C, KnOTS) as alternatives to naive averaging, and also investigate aggregation of LoRA adapters separately in the \(A\) and \(B\) factorized space. On the client side, the paper compares **hard initialization** with **regularization approaches** (parametric and function-space), and introduces a **similarity-based heuristic** to modulate regularization strength depending on how close the client model is to the global model.

---

## Strengths
- Provides a **clear and principled framing** of aggregation at both server and client levels, making the paper easy to follow and well-written.
- Systematically compares multiple merging and guidance approaches under a unified PFL setup.
- The **separate treatment of LoRA factors (\(A, B\))** is well-motivated and empirically useful.
- The **similarity-based adjustment** strategy is novel in this context.

---

## Weaknesses
- **Experimental evaluation is limited**: results are reported on only one dataset (SuperNI) and one model (LLaMA-1B). More diverse settings are needed to assess robustness.
- **Missing baselines**: no direct comparison to established personalized FL methods (e.g., Ditto, FedPer, pFedMe, FedRep). Without this, it is unclear whether the proposed strategies would outperform existing PFL approaches.
- **Communication cost**: advanced merging strategies require transmission of full model updates, which undermines one of the main benefits of LoRA in federated learning (low communication overhead).
- **Contributions are modest**: model merging for FL and regularization-based personalization are already studied. The only distinct contribution is the similarity-based heuristic, which is interesting but incremental.

---

## Overall Evaluation
The paper is **well-written and easy to follow**, and provides a structured comparison of server-side and client-side strategies for federated LoRA fine-tuning. However, the contributions are relatively modest, the experiments are weak, and the work does not offer strong novelty. It somewhat fits thematically but does not introduce anything particularly interesting.

---

**Score:**

2

**Topic Fit:**

2

---

### Official Review · Reviewer_H1mu · 2025-09-14
**Manuscript becomes more attractive if the authors' contribututions are more focused**

**Confidence:** 2

**Review:**

1. Overview

The paper, "From Aggregation to Guidance: Strategies for Personalized Federated Fine-Tuning of Foundation Models," presents a comprehensive study on improving Personalized Federated Learning (PFL) for LLMs fine-tuned with LoRA. The work approaches this by investigating strategies at two key stages: Server-side aggregation, for creating a global model, and Client-side learning guidance, for applying that model to local training. While the paper presents several findings, its broad scope can lead to ambiguity regarding its primary research focus.

2. Analysis of the Paper's Contributions

The paper's contributions can be interpreted in several ways, reflecting its multifaceted approach.

2.1 As a Comparative Study:

The paper functions as a systematic evaluation of various advanced techniques. On the server side, it benchmarks methods like TIES-Merging and Iso-C against the standard FedAvg baseline. On the client side, it compares different guidance strategies, showing that regularization-based methods outperform simple hard initialization. From this perspective, the paper serves as an empirical report, with key takeaways such as "Iso-C merging in KnOTS space... achieves the best performance" in the tested environment.

2.2 As a Proposal for a Novel Technique:

The most distinct novel contribution is the similarity-based adjustment strategy. This method addresses the stated problem that "uniform regularization across clients may be insufficient" by proposing to change the regularization coefficient, β, based on client-global model similarity. This allows clients with distinct tasks to be less constrained by the global model during local training.

3. Discussion of the Paper's Focus and Limitations

The paper's comprehensive approach, which combines an empirical survey with a novel proposal, is an interesting and valuable contribution. However, it can lead to an impression of an ambiguous primary focus. This breadth has implications for the work's overall impact and reproducibility. One issue is that the study's findings are constrained by its experimental scope. The experiments are conducted on a single, relatively small model and a single dataset with 10 clients. These factors limit the generalizability of the results to larger-scale or different PFL scenarios. The absence of a non-federated baseline also makes it difficult to assess the absolute benefit of the proposed federated methods over isolated client training.

**Score:**

3

**Topic Fit:**

2